# The development of laparoscopic skills using virtual reality simulations: A systematic review

João Victor Taba[1], Vitor Santos Cortez[1], Walter Augusto Moraes[1], Leandro Ryuchi Iuamoto[2], Wu Tu Hsing[2], Milena Oliveira Suzuki[1], Fernanda Sayuri do Nascimento[1], Leonardo Zumerkorn Pipek [1], Vitoria Carneiro de Mattos[1], Eugênia Carneiro D'Albuquerque[3], Luiz Augusto Carneiro-D'Albuquerque [4], Alberto Meyer [4]*, Wellington Andraus[4]

1 Faculty of Medicine FMUSP, University of São Paulo, São Paulo, SP, Brazil, 2 Center of Acupuncture, Department of Orthopaedics and Traumatology, University of São Paulo, São Paulo, SP, Brazil, 3 Medicine Academic -Santa Marcelina Faculty, São Paulo, SP, Brazil, 4 Department of Gastroenterology, Hospital das Clínicas, HCFMUSP, São Paulo, SP, Brazil

* alberto.meyer@usp.br

## Abstract

### Background

Teaching based on virtual reality simulators in medicine has expanded in recent years due to the limitations of more traditional methods, especially for surgical procedures such as laparoscopy.

### Purpose of review

To analyze the effects of using virtual reality simulations on the development of laparoscopic skills in medical students and physicians.

### Data sources

The literature screening was done in April 2020 through Medline (PubMed), EMBASE and Database of the National Institute of Health.

### Eligibility criteria

Randomized clinical trials that subjected medical students and physicians to training in laparoscopic skills in virtual reality simulators.

### Study appraisal

Paired reviewers independently identified 1529 articles and included 7 trials that met the eligibility criteria.

### Findings

In all studies, participants that trained in virtual simulators showed improvements in laparoscopic skills, although the articles that also had a physical model training group did not show better performance of one model compared to the other.

**Data Availability Statement:** All relevant data are within the paper and its Supporting Information files.

**Funding:** The author(s) received no specific funding for this work.

**Competing interests:** The authors have declared that no competing interests exist.

## Limitations

No article beyond 2015 met the eligibility criteria, and the analyzed simulators have different versions and models, which might impact the results.

## Conclusion

Virtual reality simulators are useful educational tools, but do not show proven significant advantages over traditional models. The lack of standardization and a scarcity of articles makes comparative analysis between simulators difficult, requiring more research in the area, according to the model suggested in this review.

## Systematic review registration number

Registered by the Prospective Register of Systematic Reviews (PROSPERO), identification code CRD42020176479.

## Introduction

In 1950 the time it took for medical knowledge to double was estimated to be 50 years, whilst in 2020 that time would be 73 days [1]. To keep up with this growth and adapt to the challenges that healthcare presents, new technologies involving both the social role of the profession and changes in the healthcare environment are considered promising complementary tools [2]. They can be used both to treat diseases and promote health (*e.g.* smokers [3] and the chronically ill [4]), as well as to help professionals with their practice and training (*e.g.* recognition of sepsis [5] and trauma screening [6]).

Among these new technologies, virtual reality (VR)—a computer simulation where the physical presence of the user is projected in a virtual environment [7]—is gaining in popularity and becoming more accessible [8–10]. Its development has been rapid, showing great application to education and training [11, 12]. Within the medical field, where its use is already widespread in procedures, diagnoses and professional training, virtual reality has great potential for expansion [13].

In the surgical context, the use of VR is highlighted in training for minimally invasive procedures, such as laparoscopy, through the use of different simulators on the market—MIST-VR, *LapSim*, *laparoscopy* VR and SINERGIA [14]. Specifically for this type of procedure, learning occurs empirically through trial and error until the technique is perfected, with the learning curve revolving around 65 procedures for laparoscopists [15]. In this way, the use of virtual reality simulators (VRS) provides safe, controlled environments, with reusable resources and with techniques that can be more easily measured when compared to practice in real models, reducing the learning curve [16].

Over the past decade, although some literature reviews have sought to analyze the use of VRS for the education and training of health professionals in a general surgical context, most do not address its use for laparoscopy specifically. In 2004, Aggarwal R. et al. [17] suggested the need to validate curricula to improve medical teaching in surgery. Willis R.E. et al. [18] noted that this technology would be more similar to a video game than to a training method.

On the other hand, most reviews that sought to analyze the VRS within the laparoscopy training field faced significant limitations [19–22]. According to Nagendran M. et al. [19], the use of VRS could decrease the surgical time of surgical trainees with laparoscopic experience—

the articles analyzed showed approximate reductions of between 30–58%. Despite the promising results, this review faced many difficulties, as all trials analysed were classified as high risk of bias, only two compared VRS with a different training method, and one of them did not fully disclose the magnitude of results. Another review by Alaker M et al. [20] also studied the effects of VRS and showed encouraging results, though the bulk of its articles focused on comparisons between VRS and no training-groups, and it did not make considerations for the costs of modern VRS and how they compare to other training methods.

For these reasons, a new systematic literature review is needed, analysing the effects of the use of virtual reality simulations on the development of laparoscopic skills in medical students and physicians.

## Objectives

Develop a systematic review of the literature to analyse the effects of using virtual reality simulations on the development of laparoscopic skills in medical students and physicians.

## Methodology

This systematic review was carried out in accordance with the items of Preferred Reports for Systematic Reviews and Protocol Meta-Analysis (PRISMA-P) [23]. This study was registered by the Prospective Register of Systematic Reviews (PROSPERO, identification code CRD42020176479) before the research was carried out.

The elaboration of the scientific question was based on the PICO strategy [24], considering: medical professionals or students (patient or problem); use of virtual reality and physical model simulations (Intervention); there is no standard comparator to be considered in this study (Control or Comparison); all outcomes available in the literature were considered in the analysis (outcome).

## Eligibility criteria

### Types of studies

The articles were selected based on their titles and abstracts according to the relevance of their data regardless of their publication status. Only clinical trials were considered.

### Types of participant

Study participants were medical students and physicians who underwent VRS training aimed at developing laparoscopic skills.

### Types of evaluation

Only physical models simulators (PMS) were considered as evaluation models, which were defined as simulations with the use of cadavers, anatomical parts, animals or a black box (trainer-boxes).

### Types of variables / Parameters analysed

Data regarding the authors, date and location (country) of the publication were collected and arranged in tables. Data were also collected regarding the number of participants analyzed in the study, sex, age, training, predominance of right handedness, previous surgical experience, training time and simulation methods used for evaluation.

## Exclusion criteria

Studies will be excluded if: (1) they have heterogeneous populations in terms of academic degree; (2) do not use a standard assessment method for the entire duration of the study, or do not have pre-assessment; (3) use VRS or augmented reality simulators as the single evaluation method or in a control group; (4) are not related to the question in the review; (5) are in a language other than english, portuguese or spanish; (6) are incomplete, unpublished or inaccessible articles to authors.

## Literature review

The survey was conducted on April 20, 2020, without language restrictions, in the Medline database (via PubMed) - www.pubmed.com, EMBASE - www.embase.com and Database of the National Institute of Health

Using the search tool, we selected MeSH terms from the most relevant publications to conduct a new search, in order to obtain articles that could be included in this systematic review.

In addition, a manual search of theses, meetings, references, study records and contact with experts in the field was carried out.

## Search strategy

The keywords were equally used in all databases, respecting their heterogeneities (for example, terms "Emtree" and terms "MeSH" were mapped in Embase and Medline, respectively).

The keywords were: "data display", "computer simulation", "learning", "education", "students, medical", "education, professional", "education, medical, continuing", "education, medical, graduate","Education, medical","internship and residency","laparoscopy".

The search strategy was: ((data display) OR (computer simulation)) AND ((learning) OR (education)) AND ((students, medical) OR (Education, Professional) OR (Education, Medical, Continuing) OR (Education, Medical, Graduate) OR (education, medical) OR (internship and residency)) AND (laparoscopy).

## Data extraction

The data for each study were extracted independently by three authors (JVT, VC and WAM). Disagreements were resolved by consensus. If no consensus was reached, a fourth author (AM) would be consulted. Data extraction was carried out using the Rayyan tool - https://rayyan.qcri.org/ [25].

All studies were analyzed based on their titles and abstracts, according to inclusion and exclusion criteria. If the eligibility criteria were met, the full text would be extracted. All studies that were eligible for qualitative analysis were described in the "Results" section.

Missing data were clarified by contacting the authors directly.

## Data validation

Three authors (JVT, VC and WAM) carried out the data validation through the discussion of the selected works. If no consensus was reached, a fourth author (AM) would be consulted.

The risk of bias for intervention-type studies was analyzed using the guidelines of the Cochrane Back Review Group (CBRG) [26].

All selected studies were considered.

## Statistical analysis

A descriptive synthesis will be produced with tables and figures and, if a number of studies with sufficient quality are available, a meta-analysis will be carried out with measures of heterogeneity and publication bias. The data will also be presented through forest-plots, according to their statistical relevance.

## Responsibility/Author contributions

All the authors involved participated in the drafting of the systematic review project, by identifying key articles, selecting keywords and writing the review project. The first author (JVT) was responsible for coordinating the group, guiding the organization of the review project, searching for articles to be reviewed and writing the text. The second (VSC) and the third author (WAM) were also responsible for the review and writing of the project, as well as for the search for articles to be reviewed. The remaining authors, in turn, were responsible for guiding and evaluating the final text.

# Results

## Research flow

The electronic search found 1904 results for the keywords used. After removing 375 duplicates, we considered 1529 potentially eligible studies. Of these, 1476 did not respect the inclusion criteria. After accessing the full text, three were excluded because they had a heterogeneous population, 24 who did not use a standard evaluation method or did not have a pre-evaluation, four because they used VRS or augmented reality simulator as a single evaluation method or in a control group, six for presenting inaccessible full text and nine for not complying with the inclusion criteria. Only seven studies were considered eligible for qualitative analysis and only one article was eligible for meta-analysis [Fig 1].

## Quality of evidence

After reading the articles included in the systematic review, the following factors were analysed to determine the level of evidence: study design and selection, detection, loss, reporting and information bias. The summary of the risk of bias analysis for each of the included articles was shown in Figs 2 and 3.

Only 2 of 7 articles had a low risk of bias by the randomization process: Ahlberg G et al. and Palter VN et al. reported clearly that the allocation process was carried out randomly and blindly [27, 28]. The other 5 were classified as having some concerns for not reporting this information.

Regarding the bias due to intended intervention (effect of adhering), 6 of the 7 articles presented low risk due to the non-applicability of this criterion to most signaling questions in this domain. Van Bruwaene S et al. was the only study that presented a high risk of bias. He reported preliminary considerations on the procedures in case of any technical failures in the implementation of the experiment. However, it cannot be said that the researchers took appropriate compensatory measures, since the establishment of a minimum training time alone does not guarantee the adequate implementation of such training for the groups [29].

In the context of intended intervention (effect of assignment) bias, no article has reached the low level of bias because the nature of a training study requires that participants and supervisors are aware of their interventions. In this context, four articles presented a high risk of bias, three due to the non-completion of all assignments by participants; and one for having an

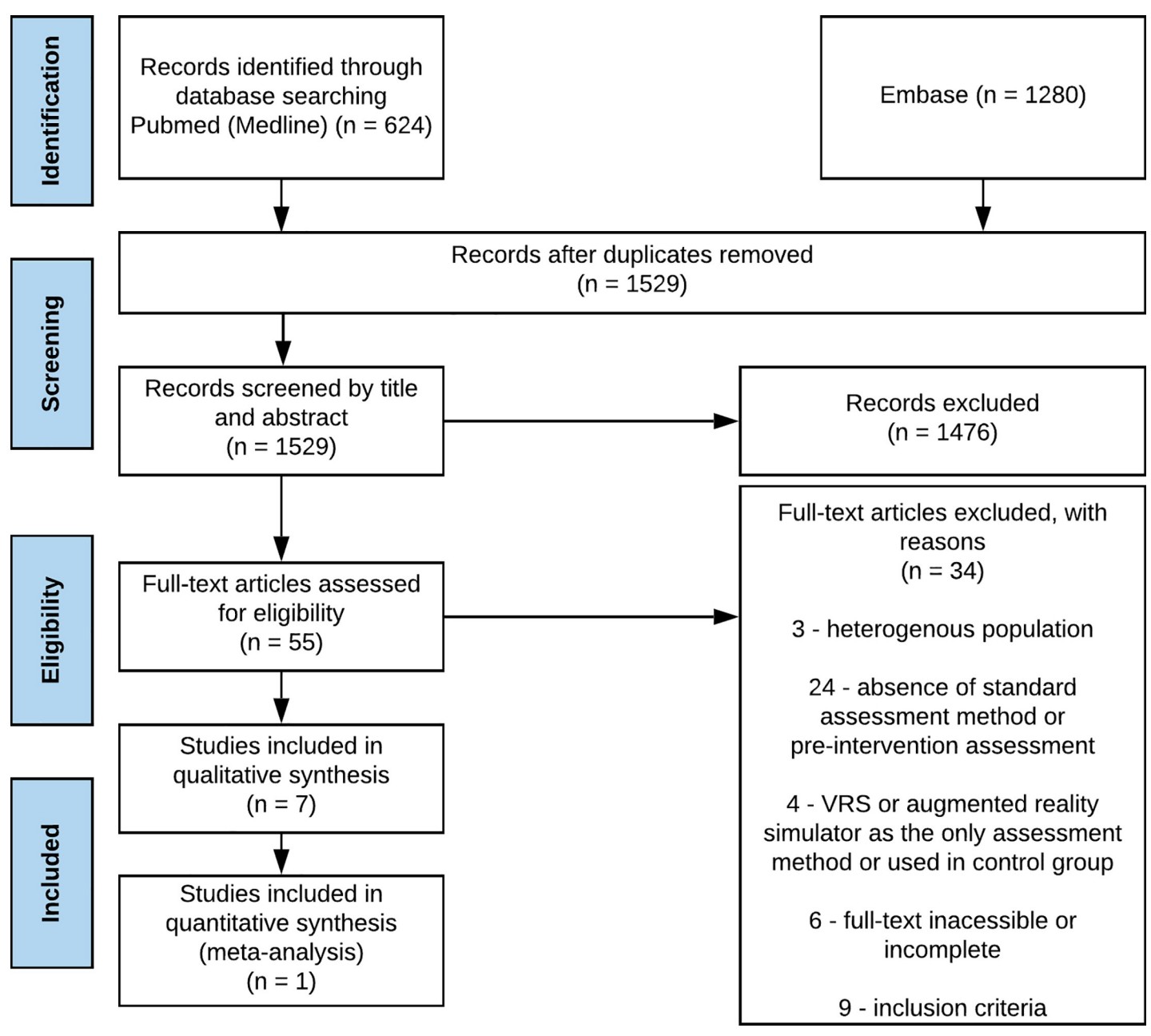

**Fig 1.**

ineligible participant. The other articles were classified as some concerns because they did not show any flaws in the analysis of the participants.

All the articles analysed showed a low missing outcome data bias. It is worth mentioning that Diesen DL et al., despite being classified under low bias, had a significant loss of data, which is more likely related to the study methodology, and not to an exclusion of unfavorable results [30]. The other authors presented the data in full of almost all study participants, without compromising the quality of the information.

For measurement of outcome bias, four of the seven articles obtained low risk of bias because they had blinded observers for the intervention and use validated measurement

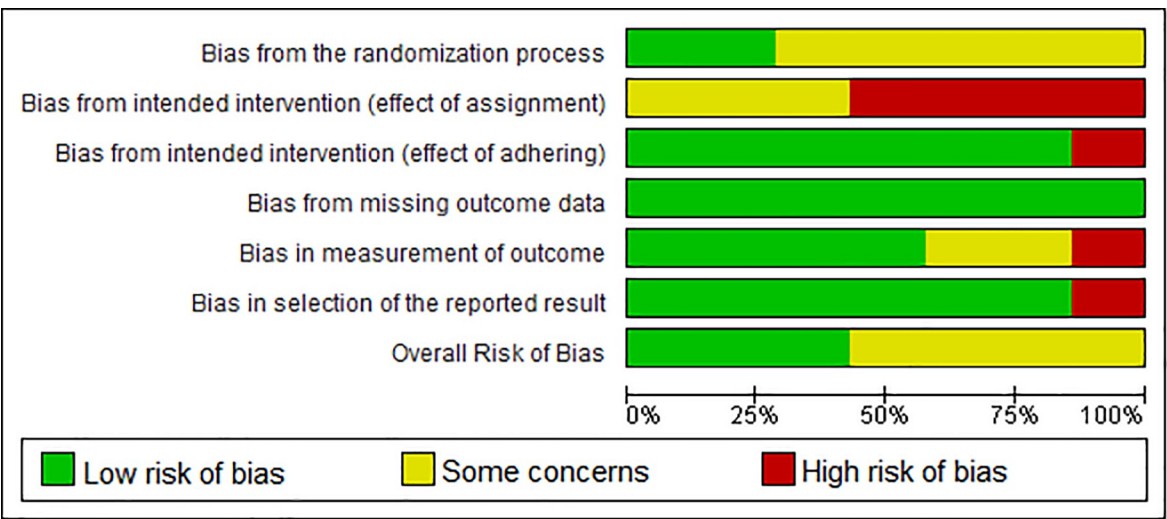

**Fig 2.**

methods that are identical between groups and with high interrater reliability. Diesen DL et al. and Munz Y et al. were classified as some concerns because they do not measure the reliability of the observers [30, 31]. Van Bruwaene S et al. had a high risk of bias due to its low interrater reliability, between 47 and 65% [29].

As for selection by the reported result bias, all studies had a low risk of bias with the exception of Ahlberg G et al., who had a high risk. This study differed from the others in that it presented different parameters in the post-intervention assessment compared to the pre-intervention. The data reported in the first evaluation included time, economy of movement and precision, whereas those in the second included only precision variables [27].

None of the seven articles analyzed presented a high risk of general bias, four of which were classified as some concerns, mainly due to the influence of the intended intervention (effect of assignment) bias and the lack of information about the randomization process.

## Study characteristics

All included studies are complete, published and have no conflict of interest. Any doubts about the available data were supplemented by contacting the respective authors. The demographic profiles are shown in Table 1; the characteristics of the methodology of the experiments are shown in Table 2; the main changes, conclusions and results are available in Tables 3–6.

Collectively the studies elected a total of 156 participants, with 40 residents and 116 medical students. It is worth considering that Diesen DL et al. partially analysed its sample, reporting only a distribution of 18 of its 23 total participants [28]. Only Ahlberg G et al., Palter VN et al. and Munz Y et al. reported a gender distribution of the sample [27, 28, 31].

Diesen DL et al., Ganai S et al. and Munz Y et al. did not perform a previous assessment of the laparoscopic skills of medical students or residents [30–32]. The remaining studies reported little or no experience in laparoscopic cholecystectomy through self-reported questionnaires or practical tests.

The training sessions in the studies lasted about 1 hour; and the time between assessments ranged from 1 week to 6 months, with Munz Y et al. and Torkington J et al. not reporting the period between assessments [31, 33].

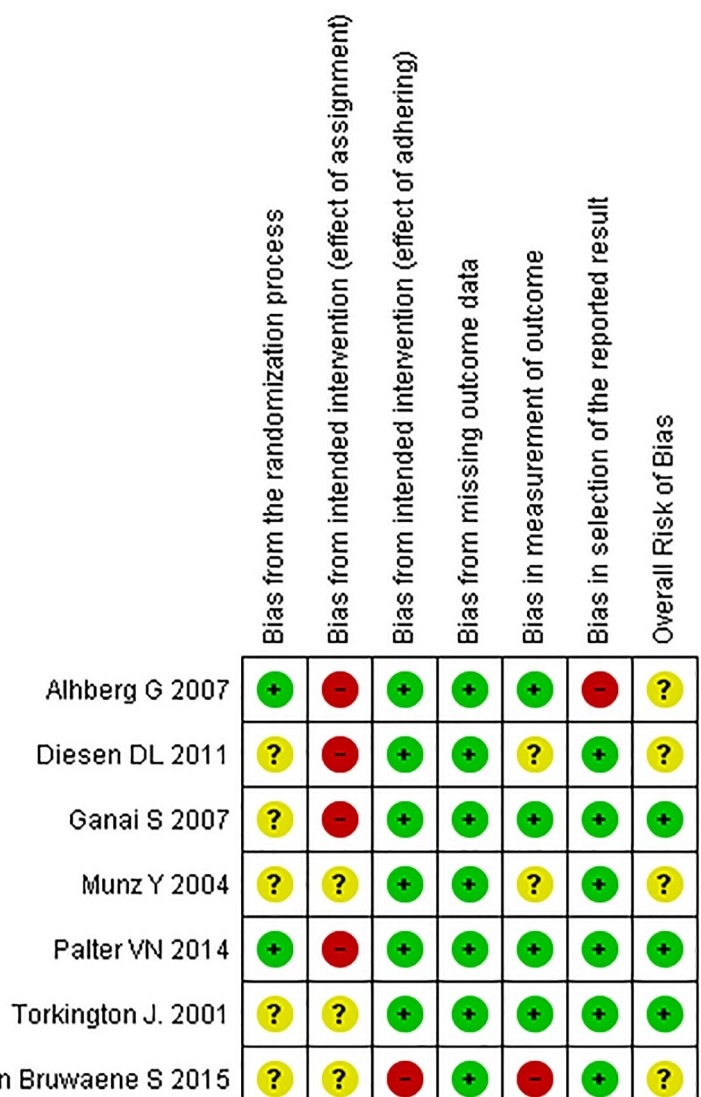

**Fig 3.**

The most prevalent VRS was the LapSim-VR-Simulator (Gothenberg, Sweden, 2008v), used in 3 of the 7 studies. Van Bruwaene S et al. used the LapMentor VR Trainer (Simbionix USA Corp) [29]; Torkington J et al. used MIST-VR (Virtual Presence, London, SE1 2NL) [33]; Ganai S et al. used Endotower (Verefi Technologies, Inc, Elizabethtown, PA) [32]; and Disen DL et al. used simulators from Medical Education Technologies, Inc. (Sarasota, Florida) and Immersion (San Jose, California) [30].

Of the 7 articles, 5 performed their assessments on PMS using laparoscopic cholecystectomy with in vivo models (3 swine and 2 humans), and Ganai S et al. only performed a telescope navigation assessment during the porcine procedure [32]. Only 2 studies carried out their assessments on PMS with non-living models, with Munz Y et al. using exercises in a water-filled glove that mimicked a gallbladder; and Torkington J et al. specific exercises in Box-trainer (BT) [33].

**Table 1. Demographic characteristics of the studies.**

| Author, publication date and country | Number of participants | Academic status | Past surgical experience |
|---|---|---|---|
| Van Bruwaene S *et al.* (2015), Belgium | Total: 30<br>CG: 10<br>IG1 (VR): 10<br>IG2 (Organ training): 10 | Medical Students | None or barely any |
| Palter VN et al. (2014), Canada | Total: 16<br>CG: 8<br>IG: 8 | Residents | <10 laparoscopic cholecystectomies |
| Diesen DL *et al.* (2011), USA | Total:18<br>IG1 (VR): 10<br>IG2 (BT): 8<br>Medical interns (first year residents):11<br>Medical students:12 | Medical Students and Residents | - |
| Ganai S *et al.* (2007), USA | Total:20<br>CG:10<br>IG:10 | Medical Students | - |
| Ahlberg G *et al.* (2007), Sweden | Total: 13<br>CG: 6<br>IG: 7 | Residents | Only assisted laparoscopic cholecystectomy<br>CG: median 18 (range 10–30)<br>IG: median 15 (range 10–25) |
| Munz Y *et al.* (2004), United Kingdom | Total: 24<br>CG:8<br>IG1 (VR): 8<br>IG2 (BT): 8 | Medical Students | - |
| Torkington J. *et al.* (2001), United Kingdom | Total: 30<br>CG: 10<br>IG1 (VR): 10<br>IG2 (Surgical Trainer): 10 | Medical Students | None or barely any |

CG = Control Group. IG = Intervention Group. VR = Virtual Reality. BT = Box-trainer.

Regarding the different parameters analyzed, 4 of the 7 articles measured the total time of the procedures during the assessments, with an improvement in the average time of the VRS groups in relation to the control groups. The study with the most expressive results was that of Ahlberg G et al., who reported that the intervention group performed the surgery 58% faster when compared to the control group [27]. On the other hand, that of Van Bruwaene S et al. showed a difference of just over 5% between these groups, with a reduction of 21.3% within the VRS group and 17% for the control group [29].

The economy of movement was explored by 2 articles, which analyzed different variables. Ahlberg G et al. considered the distance covered in meters and angular movement of the instruments, whereas Torkington et al. assessed the number of movements per hand, hand speed and variation in hand distance during the experiment. In both studies, improvements in these criteria do not appear to be clear or significant [27, 33].

In the precision parameters, the total number of errors was analyzed by 2 articles, both of which showed a significant improvement in the average of errors, especially in Ganai S et al., whose number of errors in the VRS group was less than half of the control group [32]. On the other hand, although Ahlberg G et al. searched for total errors only in the post-assessment, the drop rates for this parameter are similar [27].

Palter VN et al. and Diesen DL et al. evaluated their assessments through scores. Palter VN et al. used the OSATS scale, obtaining significant improvement in its indexes for the VRS group in relation to the control (p <0.03). Diesen DL et al. developed its own scale for the study, obtaining better results in general for the BT group in relation to VRS, with emphasis

**Table 2. Study methods.**

| Author, publication date and country | VRS utilized by intervention group | Instruments utilized by other groups | Method of assessment before and after training | Tasks applied in training | Main parameters | Time in training | Time between assessments |
|---|---|---|---|---|---|---|---|
| Van Bruwaene S *et al.* (2015), Belgium | LapMentor VR Trainer (Simbionix USA Corp) | CG: No training IG2: Organs (dissections) | Laparoscopic cholecystectomy on a live porcine model | Complete Cholecystectomy skills: 1. Bean Drop 2. Rope Pass 3. Checkerboard 4. Laparoscopic skills testing and training (LASTT) 5. Suturing | **A. Time:** Time for each task **D. Others:** Overall expert's grade | IG1 (VR): 5h sessions (10 sessions) IG2 (Organ): 1h daily (10 days) | 1 week |
| Palter VN et al. (2014), Canada | LapSim VR Simulator (Gothenberg, Sweden, 2008 version) | CG: No training | Laparoscopic cholecystectomy on a patient | 1. Instrument navigation 2. Grasping 3. Cutting 4. Clipping 5. Lifting and Grasping | **B. Economy of movement:** Economy of movement (time and motion) **C. Precision:** 1. Respect for tissue 2. Precision of operative technique **D. Others:** 1. Instrument handling 2. Knowledge of instruments 3. Use of assistants 4. Flow of operation and forward planning 5. Knowledge of specific procedure | 1h sessions | Median of 18 days (range 14–36) |
| Diesen DL *et al.* (2011), USA | Computer simulation | IG2: BT | Laparoscopy on a porcine model | 1. 30˚ camera navigation 2. Clipping and electrocautery 3. Knot-tying 4. 2-eye-hand coordination exercises | **B. Economy of movement:** Economy of movements **D. Others:** 1. Perceptual ability 2. Scope orientation 3. Appropriate control of needles and instrument handling 4. Appropriate clipping and eletrocautery | 6 months | 6 months |
| Ganai S *et al.* (2007), USA | EndoTower (Verefi Technologies, Inc, Elizabethtown, PA) | CG: No training | Telescope navigational assessment on a porcine model | Angled-telescope navigation | **A. Time:** Total time **C. Precision:** 1. Horizon and total errors 2. Instrument collisions and scope smudges **D. Others:** 1. Visualization of the object of interest centered on the monitor and proper scope orientation | 1 hour sessions | 3 to 4 weeks |

(*Continued*)

**Table 2.** (Continued)

| Author, publication date and country | VRS utilized by intervention group | Instruments utilized by other groups | Method of assessment before and after training | Tasks applied in training | Main parameters | Time in training | Time between assessments |
|---|---|---|---|---|---|---|---|
| Ahlberg G *et al.* (2007), Sweden | LapSim VR Simulator (Surgical Science Inc., Gothenburg, Sweden) | CG: No training | Laparoscopic cholecystectomy on a patient | 1. Suturing with and without easy grip function 2. Lifting 3. Grasping 4. Clipping 5. Ultrasonographic dissection with both hands | **A. Time:** Total time **B. Economy of movement:** 1. Instrument path length 2. Instrument angular length **C. Precision:** 1. Tissue damage 2. Maximum and minimum damage 3. Errors | 1 week | Within 6 months |
| Munz Y *et al.* (2004), United Kingdom | LapSim VR Simulator (Surgical Science Inc., Gothenburg, Sweden) | CG: No training IG2: BT | BT cutting and clipping task | 1. Grasping 2. Cutting | **A. Time:** Total time for each task **B. Economy of movement:** 2. Total number of hand movements made 3. Total distance traveled for each hand 4. Economy of hand movement **C. Precision:** 5. Number of errors | 30 min sessions per week (3 sessions) | - |
| Torkington J. *et al.* (2001), United Kingdom | MIST-VR (Virtual Presence, London, SE1 2NL) | CG: No training IG2: Surgical Trainer | BT grasping and cutting suture task | 1. Instrument Navigation 2. Coordination 3. Grasping 4. Cutting 5. Precision and Speed | **A. Time:** Total time **B. Economy of movement:** 1. Number of movements made 2. Distance traveled by each instrument 3. Speed of travel | 1 hour sessions | - |

CG = Control Group. IG = Intervention Group. VR = Virtual Reality. BT = Box-trainer. MIST-VR = Minimally Invasive Surgery Trainer-Virtual Reality.

on the instrument handling exercises, especially in needle transfer (p <0.0002) and in camera navigation (p <0.006) [28, 30].

No study reported significant differences between groups in the pre-intervention assessment. In the post-intervention assessment, All seven of the assessments showed that VRS produced an improvement in the laparoscopic skills of the participants, although Munz Y et al. and Torkington J et al. have not found differences in performance between VRS or PMS training that would justify the superiority of one method over the other [31, 33]. Van Bruwaene S et al. was the only one whose group with training in PMS obtained better results in all parameters in relation to the VRS group, also expressing the need for further studies to define the quality of teaching by VRS [29].

## Discussion

VRS-based teaching is expanding in medicine due to the limitations of more traditional methods, especially for surgical procedures [34, 35]. There are already several areas, such as neurosurgery [36], ophthalmic surgery [37] and digestive endoscopy [38], considering the

**Table 3. Main changes and conclusions of the studies.**

| Author, publication date and country | Pre-intervention assessment main results | Post-intervention assessment main results | Study conclusions |
|---|---|---|---|
| Van Bruwaene S *et al.* (2015), Belgium | **A. Intergroup:** No significant differences were found between groups | **A. Intergroup:** 1. IG2 (Organ) had the shortest time and IG1 (VR) improved compared to CG 2. IG2 (Organ) had better overall quality score and IG1 (VR) improved compared to CG **B. Intragroup:** Both IGs showed improvement between assessments | For trainees who are proficient in basic laparoscopic skills, the efficacy of the VRS training model remains to be proven |
| Palter VN et al. (2014), Canada | **A. Intergroup:** Both groups presented similar OSATS scores in OR | **A. Intergroup:** IG exhibited superior OSATS scores than the CG in OR | Deliberate individualized practice on VRS could improve technical performance in the operating room. This could mean that implementing a simulation-based curricula in residency training programs could lead to positive results |
| Diesen DL *et al.* (2011), USA | **A. Intergroup:** No significant differences were found between groups | **A. Intergroup:** No significant difference between training groups **B. Intragroup:** Overall improvement in scores after training | BT and VRS are equally effective means of teaching laparoscopic skills to novice learners |
| Ganai S *et al.* (2007), USA | **A. Intergroup:** No significant differences were found between groups | **A. Intergroup:** 1. IG had a more notable improvement in object visualization, scope orientation, horizon scores, number of smudges and collisions and errors **B. Intragroup:** 1. Both groups showed improvement in time 2. Both groups showed improvement in scope orientation scores between tests | VRS can be used to improve operative surgical skill |
| Ahlberg G *et al.* (2007), Sweden | **A. Intergroup:** No significant differences were found between groups | **A. Intergroup:** CG presented a variation in total errors 8 times higher than IG | VRS could improve the initial learning curve in laparoscopic procedures, such as laparoscopic cholecystectomy. The LapSim Simulator should be used to train new laparoscopists until they reach a proficiency level. |
| Munz Y *et al.* (2004), United Kingdom | **A. Intergroup:** No significant differences were found between groups | **A. Intergroup:** IG1 (VR) and IG2 (BT) performed better than CG but there were no significant differences between one another **B. Intragroup:** Both IGs showed overall improvement in scores after training | LapSim can be used to teach skills that are transferable to real laparoscopic tasks, but it appears that there are no advantages to using VR over BT and vice versa |
| Torkington J. *et al.* (2001), United Kingdom | **A. Intergroup:** No significant differences were found between groups | **A. Intergroup:** 1. Left-hand performance: both IGs were less economical in the number of movements and tended to move at greater speeds compared with CG 2. Right-hand performance: both IGs were more economical in the number of movements; and there were no significant change in time completion compared with the CG **B. Intragroup:** 1. Both IGs showed worse left-hand performance 2. Both IGs showed better right-hand performance | MIST-VR can be used by novices to transfer skills to simple real tasks and its results are similar when compared with conventional training |

CG = Control Group. IG = Intervention Group. VR = Virtual Reality. BT = Box-trainer. OR = Operation room. VRS = Virtual reality simulator. MIST-VR = Minimally Invasive Surgery Trainer-Virtual Reality. OSATS = Objective Structured Assessment of Technical Skills.

implementation of this technology in their training curricula. In this scenario, systematic reviews were produced to analyse the relation between the use of VRS and the learning of surgical techniques, considering the expansion of this technology in medical teaching curricula [39–43].

The field of laparoscopic surgery also follows this trend, with conclusions that vary depending on the studies. In general terms, reviews conclude that VRS is a method with the potential

**Table 4. Main study time changes.**

| Measured time parameters | | Van Bruwaene S *et al.* (2015), Belgium | | | Ganai S *et al.* (2007), USA | | | Ahlberg G *et al.* (2007), Sweden | | | Torkington J. *et al.* (2001), United Kingdom | | |
|---|---|---|---|---|---|---|---|---|---|---|---|---|---|
| | | Mean value | SD | P value | Mean value | SD | P value | Mean value | SD | P value | Mean value | SD | P value |
| Pre-assessment | Total time (s) | CG: 2820 IG1(VR): 2820 IG2 (Organ): 2760 | IQR: CG: 1260 IG1(VR): 600 IG2 (Organ): 660 | 0.642' | CG: 426 IG: 388 | 95% CI: CG: 360–492 IG: 321–456 | - | CG: 114 IG: 110 | Range: CG: 73.4–182.4 IG: 61.9–150.9 | - | - | | |
| Post-assessment | Total time (s) | CG: 2340 IG1(VR): 2220 IG2 (Organ): 1800 | IQR: CG: 720 IG1(VR): 1320 IG2 (Organ): 120 | 0.046' | CG: 296 IG: 260 | 95% CI: CG: 238–354 IG: 194–326 | < 0.05 | 58% longer in CG compared with IG | - | 0.586' | - | | |
| Comparison between assessments | Total time variation (s) | - | | | - | | | - | | | CG: -9.4 IG1(VR): -23.7 IG2 (Surgical Trainer): -7.0 | CG: 12 IG1(VR): 7.8 IG2 (Surgical Trainer): 8.8 | One-way ANOVA analysis of all groups: 0.43 |

CG = Control Group. IG = Intervention Group. VR = Virtual Reality. IQR = Interquartile range. CI = Confidence Interval. ANOVA = Analysis of Variance.

to develop varied surgical skills. However, most articles only compare VRS with untrained groups; whilst articles that use comparative groups with other methods show mixed results [19, 20].

Despite recommending the incorporation of VRS in laparoscopic surgical training curricula, Alaker M et al. did not observe any statistically significant difference between VRS and BT groups regarding time and score. In addition, their comparison of 'virtual reality vs box and video trainers combined' did also fail to find statistically significant differences between VRS and these more traditional methods [20]. Another review, by Nagendram et al. found that operative performance was significantly better in the VRS group in comparison to the BT group, but this was limited to only two articles, of which, one did not fully disclose the magnitude of the difference or other quantitative results [19]. Thus, there is not enough evidence to justify the use of this new technology in place of more traditional training in PMS.

In this systematic review, 6 of the 7 articles analysed compared the performance between a VRS group and a control group, 3 of which also used a third group in PMS. All observed an improvement of the new technology in relation to the control, evidencing the VRS as a viable alternative for the teaching [27–29, 31–33]. However, part of our results were similar to those of Alaker M et al. [20]: in all the articles we analysed that compare VRS with PMS, the use of technology has not shown significant gain [29–31, 33].

One explanation for this outcome lies in the sample limitations of these studies, whose population includes medical students and residents. There is a significant difference between the level of surgical and clinical experience between these two population types and, in this sense, medical students may not fully benefit from the training they have undergone [29–31, 33]. In addition, the sample size of these studies, between 18 to 30 participants [29, 30, 33], may not

**Table 5. Main changes in movement economics of the studies.**

| Measured economy of movement parameters | | Ahlberg G *et al*. (2007), Sweden | | | Torkington J. *et al*. (2001), United Kingdom | | |
|---|---|---|---|---|---|---|---|
| | | Mean value | SD | P value | Mean value | SD | P value |
| **Pre-assessment** | Left instrument path length (m) | CG: 1.4 IG: 1.3 | Range: CG: 1.1–1.7 IG: 1.2–1.6 | - | - | | |
| | Left instrument angular path (˚) | CG: 314.7 IG: 317.8 | Range: CG: 276.4–470.2 IG: 253.2–397.9 | - | | | |
| | Right instrument path length (m) | CG: 1.2 IG: 1.2 | Range: CG: 1.0–1.4 IG: 1.0–1.4 | - | | | |
| | Right instrument angular path (˚) | CG: 274.9 IG: 290.8 | Range: CG: 238.5–358.9 IG: 240–365.8 | - | | | |
| **Comparison between assessments** | Number of movements variation for left hand (#) | - | | | CG: -11.0 IG1(VR): 0.9 IG2(Surgical Trainer): 10.0 | CG: 5.6 IG1(VR): 2.9 IG2(Surgical Trainer): 6.5 | One-way ANOVA analysis of all groups: 0.03 (0.02a) |
| | Speed of travel variation for left hand(cm/sec) | | | | CG: -0.2 IG1(VR): 0.5 IG2(Surgical Trainer): 0.4 | CG: 0.3 IG1(VR):0.2 IG2(Surgical Trainer): 0.2 | One-way ANOVA analysis of all groups: 0.04 (0.01a) |
| | Distance variation for left hand (cm) | | | | CG: -34.8 IG1(VR): 9.8 IG2(Surgical Trainer): 31.8 | CG: 23.7 IG1(VR): 11.6 IG2(Surgical Trainer): 2.8 | One-way ANOVA analysis of all groups: 0.11 |
| | Number of movements variation for right hand (#) | | | | CG: 3.7 IG1(VR): -17.9 IG2(Surgical Trainer): -7.0 | CG: 7.1 IG1(VR): 6.4 IG2(Surgical Trainer): 3.5 | One-way ANOVA analysis of all groups: 0.05 (0.03a) |
| | Speed of travel variation for right hand(cm/sec) | | | | CG: 0.1 IG1(VR): 0.2 IG2(Surgical Trainer): 0.0 | CG: 0.1 IG1(VR): 0.2 IG2(Surgical Trainer): 0.3 | One-way ANOVA analysis of all groups: 0.82 |
| | Distance variation for right hand (cm) | | | | CG: 5.2 IG1(VR): -45.6 IG2(Surgical Trainer): -7.3 | CG: 25.8 IG1(VR): 25.6 IG2(Surgical Trainer): 16.9 | One-way ANOVA analysis of all groups: 0.29 |

CG = Control Group. IG = Intervention Group. VR = Virtual Reality. ANOVA = Analysis of Variance.

have been enough to achieve an adequate population representation. Thus, the results obtained in VRS may have been underestimated.

Another factor that can contribute to better performance results in PMS when compared to VRS is the tool used to carry out the assessments. Of the 4 articles, 2 (Munz Y et al. And Torkington J et al.) performed their pre- and post-training measurements on box-trainers [31, 33]. This can give an advantage to groups that trained in PMS over VRS groups due to the similar nature in training and assessment.

Due to the very nature of digital simulations, with software and engines that are constantly updating, it is difficult for the literature to follow and be able to produce studies that are consistent with the current situation of VRS. The studies analysed in this systematic review are

**Table 6. Main changes in study accuracy.**

| Measured precision parameters | | Ganai S *et al.* (2007), USA | | | Ahlberg G *et al.* (2007), Sweden | | |
|---|---|---|---|---|---|---|---|
| | | Mean value | SD | P value | Mean value | SD | P value |
| **Pre-assessment** | Horizon error (#) | CG: 4.4<br>IG: 3.8 | 95% CI:<br>CG:(2.4–6.4)<br>IG:(2.0–5.6) | - | - | | |
| | Instrument collisions (#) | CG: 7.1'<br>VR: 5.3' | CG:(2.5–11.7)<br>VR: (2.3–8.3) | - | | | |
| | Scope smudges (#) | CG:2.1'<br>VR: 2.2' | CG:(0.3–3.9)<br>VR: (0.4–4.0) | - | | | |
| | Total errors (#) | CG:13,6<br>VR:11.3' | CG:(7.4–19.8)<br>VR: (6.9–15.7) | - | | | |
| | Tissue damage (#) | - | | | CG: 4<br>IG: 2 | Range:<br>CG: 2–9<br>IG: 0–3 | - |
| | Maximum damage (mm) | | | | CG: 5.2<br>IG: 4.7 | Range:<br>CG: 0.7–15.9<br>IG: 0–7.4 | - |
| **Post-assessment** | Horizon error (#) | CG:<br>IG: 1.3' | CG: (0.9–4.9)<br>IG:(0.7–1.9) | - | - | | |
| | Instrument collisions (#) | CG:3.5'<br>IG: 2.3 | CG:(1.4–5.7)<br>IG:(0.1–4.5) | - | | | |
| | Scope smudges (#) | CG:2.4'<br>IG: 0.4' | CG:(0.4–4.2)<br>IG:(0–1.0) | - | | | |
| | Total errors (#) | CG:8.8'<br>IG: 3.9' | CG:(1.3–6.5)<br>IG:(3.8–13.8) | - | CG: 86.2<br>IG:28.4 | 95% CI:<br>CG: 58.18–114.12<br>IG: 23.51–33.32<br>Variance:<br>CG: 916.68<br>IG: 118.69 | 0.0037' |
| | Exposure errors (#) | | | | CG: 53.4<br>IG:15.0 | 95% CI:<br>CG: 16.7–90.13<br>IG: 11.16–18.79<br>Variance:<br>CG: 623.31<br>IG: 68.44 | 0.0402' |
| | Clipping and tissue division errors (#) | | | | CG: 7.1<br>IG:1.9 | 95% CI:<br>CG: 3.95–10.25<br>IG: 0.93–2.87<br>Variance:<br>CG: 41.11<br>IG: 5.57 | 0.008' |
| | Dissection errors (#) | | | | CG: 29.5<br>IG: 11.5 | 95% CI:<br>CG: 13.99–45.01<br>IG: 8.82–14.08<br>Variance:<br>CG: 61.5<br>IG: 28.77 | 0.031' |
| **Comparison between assessments** | Horizon error (#) | - | | < 0.05 | - | | |
| | Instrument collisions (#) | | | 0.06 | | | |
| | Scope smudges (#) | | | < 0.051' | | | |
| | Total error score (#) | | | < 0.05 | | | |

CG = Control Group. IG = Intervention Group. CI = Confidence Interval.

between 5 and 19 years old and do not use equipment corresponding to the technological advances of their respective times. Palter VN et al., For example, published their work in 2014, but used a 2008 model (LapSim VR Simulator—Gothenberg, Sweden) [28]. Ahlberg G et al.

(2007) and Munz Y et al. (2004) also used this same simulator, but in even older and different versions [27, 31]. Thus, there is a great temporal difference between the development of a VRS, its validation and the review of its results in the literature.

Due to this constant updating of technology, the need to repeatedly carry out new studies to keep up to date with newer versions of VRS can bring high costs to the research team, since the equipment is usually expensive and the production of RCTs is time consuming. Speich B et al. compared the average production cost of RCTs between 2012 and 2016, concluding that, although the value may fluctuate depending on the scope of the project and study design, there was an average cost of USD 72,000.00 in preparing a survey in both times [44].

In this review, of the 7 articles analysed, 4 use VRS LapSim or LapMentor in their methodologies. Currently, the purchase price of this equipment is approximately USD 70,000.00 and USD 84,000.00, respectively, excluding USD 15,000.00 for additional modules (2018 versions) [45]. Therefore, the acquisition of only a single piece of equipment is enough to more than double the average costs of research [44], contributing to the scarcity of literature to compare this technology with PMS methods, since it is not attractive for a research group to constantly produce such expensive update articles.

In contrast to the high prices of VRS, traditional PMS, especially box-trainers, are much more accessible for acquisition and use in training healthcare professionals. A complete box-trainer, including camera and equipment, costs between USD 1,000 and USD 6,000.00 [46], up to 84 times cheaper than current VRS. It should be noted that the VRS could save significant costs for the institution by reducing the need for proctors and replenishment of animal and synthetic material. Though they have other maintenance needs typical of any software/hardware, and some demand subscriptions or the continued purchase of new modules, effectively nullifying this effect. Nevertheless, there is still a huge disparity in costs that may justify the fear of investing in VRS for medical curricula. Especially considering the absence of solid evidence about the advantages of the new technology in relation to regular methods, which was verified by 4 of our 7 articles [29–31, 33].

## Study limitations and methodologies

The most common limitation reported in the studies analyzed was the small sample size [Table 7], with an average of approximately 22.3 participants per study and 9.2 per group. Ergo, it may have decreased the statistical power of the studies and the results obtained may not be representative enough of the population. In addition, of the 7 studies, 4 used medical students and 2 used residents. Diesen DL et al. was the only one that used a mixed training sample [30]. Although not reported by the author, this population heterogeneity is a potential confounder factor in the study.

According to Van Bruwaene et al., The use of medical students in this type of training may not represent reliable results, given the lack of experience in relation to residents [29]. In this sense, residents would be a more suitable population for studies on surgical training. On the other hand, we could consider that students represent a more adequate sample due to their lesser degree of contact with laparoscopy and, therefore, would be less biased than residents, who are more exposed to practices outside the study. However, the current literature does not present significant information about the advantages of one group in relation to another.

We grouped the performance parameters of the participants into three main categories: time, economy of movement and precision. The time item was exposed quantitatively by 4 of the 7 articles [27, 29, 32, 33], while the other categories were only addressed by 2 articles each [27, 32, 33]. Thus, despite comparing the performance of VRS groups with PMS groups, the analysed articles show great methodological differences between them. This made it difficult to

**Table 7. Study limitations reported.**

| Author, publication date and country | Reported limitations |
|---|---|
| Van Bruwaene S *et al.* (2015), Belgium | 1. Small sample size |
| | 2. Medical students without clinical or surgical experience compared with the residents, might have had insufficient knowledge to fully profit from the training and its effects might have been underestimated |
| | 3. Hard to verify equal amount of training in the organ and VR training group as the first one was restricted by time and the second by proficiency parameters |
| | 4. Interrater (between raters) reliability was low |
| Palter VN *et al.* (2014), Canada | Using different supervisors in the OR |
| Diesen DL *et al.* (2011), USA | 1. Small sample size |
| | 2. Did not use the most recently released software |
| Ganai S *et al.* (2007), USA | 1. Small sample size |
| | 2. Absence of specific training for the residents (CG) |
| Ahlberg G *et al.* (2007), Sweden | 1. Small sample size |
| | 2. Using different supervisors in the OR |
| Munz Y *et al.* (2004), United Kingdom | IG2 (BT) could have advantage over IG1 (VR) by training with real laparoscopic instruments |
| Torkington J. *et al.* (2001), United Kingdom | Short training time to achieve both hands proficiency |

CG = Control Group. IG = Intervention Group. OR = Operation room. VR = Virtual reality. BT = Box-trainer.

compare the results for this review, making it impossible to develop a meta-analysis based on the articles.

Some studies have presented different methodological elements. Torkington et al. presented its data through the variation between the initial and final results, which would be more appropriate for a statistical analysis [33]. In addition, the use of validated and standardized scores could be a way to circumvent the heterogeneity of parameters, as was the case with Palter VN et al. when using OSATS [28]. Although Diesen DL et al. adopted a score in his study, it was personalized and has no widespread use in the literature [30].

Among the limitations of our systematic review, our results might have been influenced due to the use of different models and versions of VRS between studies, since the analysed articles date from 2001 to 2015. No more recent articles were found that were eligible during the screening of literature.

To better assess the use of VRS in the development of laparoscopic skills, we propose the use of RCTs that use homogeneous samples, submitted to equal assessments before and after training, which do not favour one group over the other, either by the nature of the simulator or the type of task. We suggest that they report in more detail the training process and time, as well as the participants' history of laparoscopic experience. The parameters evaluated in the assessments must be standardized so that it is possible to build a comparative analysis between the different studies. This can be done with scores validated by the literature, containing variables such as total time, number of errors and distances covered by the instruments.

## Conclusions

Concomitant to recent advances in the area, VRS was described by the 7 articles analysed in this review as an effective teaching method. However, no study has been able to point out a significant advantage of VRS when compared to PMS, although the price difference between simulators can be as high as 84 times. Nonetheless, there is a lack of standardization and a scarcity

of articles in the literature, hindering the comparative analysis of the performance in VRS in relation to PMS, therefore, more research in this area is necessary.

## Supporting information

**S1 Checklist. PRISMA 2009 checklist.**
(DOC)

## Author Contributions

**Conceptualization:** João Victor Taba, Vitor Santos Cortez, Milena Oliveira Suzuki, Fernanda Sayuri do Nascimento, Alberto Meyer.

**Data curation:** João Victor Taba, Vitor Santos Cortez.

**Formal analysis:** João Victor Taba, Vitor Santos Cortez, Walter Augusto Moraes.

**Investigation:** João Victor Taba, Vitor Santos Cortez, Leandro Ryuchi Iuamoto.

**Methodology:** João Victor Taba, Vitor Santos Cortez, Walter Augusto Moraes, Wu Tu Hsing, Milena Oliveira Suzuki, Fernanda Sayuri do Nascimento, Alberto Meyer.

**Project administration:** Wu Tu Hsing, Milena Oliveira Suzuki, Fernanda Sayuri do Nascimento, Luiz Augusto Carneiro-D'Albuquerque, Alberto Meyer, Wellington Andraus.

**Supervision:** Milena Oliveira Suzuki, Fernanda Sayuri do Nascimento, Eugênia Carneiro D'Albuquerque, Luiz Augusto Carneiro-D'Albuquerque, Alberto Meyer, Wellington Andraus.

**Validation:** Leonardo Zumerkorn Pipek, Vitoria Carneiro de Mattos, Eugênia Carneiro D'Albuquerque.

**Visualization:** Leandro Ryuchi Iuamoto, Leonardo Zumerkorn Pipek.

**Writing – original draft:** João Victor Taba, Vitor Santos Cortez, Walter Augusto Moraes, Leandro Ryuchi Iuamoto.

**Writing – review & editing:** Leonardo Zumerkorn Pipek, Vitoria Carneiro de Mattos, Eugênia Carneiro D'Albuquerque, Alberto Meyer.

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
