## [Decision Letter · Decision Letter 0]

1 Mar 2021

PONE-D-20-37286

The development of laparoscopic skills using virtual reality simulations: a systematic review

PLOS ONE

Dear Dr. Meyer,

Thank you for submitting your manuscript to PLOS ONE. After careful consideration, we feel that it has merit but does not fully meet PLOS ONE’s publication criteria as it currently stands. Therefore, we invite you to submit a revised version of the manuscript that addresses the points raised during the review process. Please address the concerns of the expert reviewers, especially the comparison with the previous surveys and meta-analyses and show the differences and discrepancies.

We look forward to receiving your revised manuscript.

Kind regards,

Mohammed Saqr, Ph.D

Academic Editor

PLOS ONE

Journal Requirements:

2. Please describe the data extraction methods in more detail. We would expect to see reporting of the specific information extracted from the manuscripts.

3. Please include your tables as part of your main manuscript and remove the individual files. Please note that supplementary tables (should remain/ be uploaded) as separate "supporting information" files

Reviewers' comments:

Reviewer's Responses to Questions

**Comments to the Author**

1. Is the manuscript technically sound, and do the data support the conclusions?

Reviewer #1: Partly

Reviewer #2: Yes

2. Has the statistical analysis been performed appropriately and rigorously? 

Reviewer #1: N/A

Reviewer #2: Yes

3. Have the authors made all data underlying the findings in their manuscript fully available?

Reviewer #1: Yes

Reviewer #2: Yes

4. Is the manuscript presented in an intelligible fashion and written in standard English?

Reviewer #1: Yes

Reviewer #2: Yes

5. Review Comments to the Author

Reviewer #1: This systematic review article summarizes the results on the use of virtual reality simulator for the development of laparoscopic skills. A total of 1529 articles were collected and out of which 7 were selected after the exclusion criteria. The authors conclude that based on the seven articles, that virtual reality simulator doesn’t provide significant advantage over traditional training .

Training in laparoscopic surgery can vary from basic to advanced procedural skills. Moreover, virtual reality simulators are not uniform with some are designed to teach basic skills and some has both basic as well as advanced skills. The metrics used for assessing can also vary significantly between the simulators. Not all simulators have force feedback. Moreover, fidelity of these simulators is also very important (ie., whether they have undergone construct validity). These issues greatly limit the usefulness of metal analysis since the number of articles were going to be always very few.

The authors have summarized the results from the 7 papers but my concern is that the conclusion is severely limited by the factors mentioned above. Unlike the authors mentioned in the paper, there has been a review published on the virtual reality training for laparoscopic surgery (Alaker M, Wynn GR, Arulampalam T. Virtual reality training in laparoscopic surgery: A systematic review & meta-analysis. Int J Surg. 2016 May;29:85-94.). Also, the authors have cited the paper by Nagendran et al., which does a meta analysis of virtual reality training in laparoscopic surgery with focus on randomized clinical trials with primary and secondary outcomes but failed to elaborate on it. In view of the two review papers that has been published and the current review paper which by exclusion criteria doesn’t include papers beyond 2015, the real impact of the metal analysis is minimal. At the least, the authors should compare the conclusions from these two meta analysis to their conclusions and explain clearly any discrepancies.

The authors mention that in the article by Diesen et al., they didn’t report the VRS used in the study, but they did mention it in the discussion section (3rd paragraph). They have used both the Medical Education Technologies Inc. laparoscopic simulator as well as the Immersion virtual reality laparoscopic simulator. One would expect a thorough review of the paper especially for meta analysis. This needs to be fixed if the authors decided to revise the paper.

Reference 13 is incomplete and is also listed as part of reference 12.

My overall recommendation is reject. If authors substantially revise and compare to existing literature reviews on VRS for laparoscopic surgery, it may be a useful contribution.

Reviewer #2: I believe that this is an important piece. It is absolutely shocking that after 15 years of marketing and pushing for laparoscopic simulation the most we can find in the literature at this point to support its use is 7 papers. The number of poorly constructed C and D level publications in this space is simply incredible. The sections of the paper breaking down these shortcomings is a call-to-arms for those in this space to structure their research correctly and to power the studies appropriately. Thank you. There are a few grammatical issues in the paper that should be reviewed; a number of past/present tense inconsistencies as well as some simple capitalizations in the middle of sentences, etc.

6. PLOS authors have the option to publish the peer review history of their article (what does this mean?). If published, this will include your full peer review and any attached files.

Reviewer #1: No

Reviewer #2: **Yes: **Juan C Cendan, MD, FACS

---

## [Author Response · Author response to Decision Letter 0]

4 Mar 2021

Dear reviewers,

Thank you for inviting us to submit a revised draft of our manuscript.

We also appreciate the time and effort you and each of the reviewers have dedicated to providing insightful feedback on ways to strengthen our paper. Thus, it is with great pleasure that we resubmit our article for further consideration. We have incorporated changes that reflect the detailed suggestions you have graciously provided. We also hope that our edits and the responses we provide below satisfactorily address all of the issues and concerns you and the reviewers have noted.

Reviewer #1: This systematic review article summarizes the results on the use of virtual reality simulator for the development of laparoscopic skills. A total of 1529 articles were collected and out of which 7 were selected after the exclusion criteria. The authors conclude that based on the seven articles, that virtual reality simulator doesn’t provide significant advantage over traditional training .

Training in laparoscopic surgery can vary from basic to advanced procedural skills. Moreover, virtual reality simulators are not uniform with some are designed to teach basic skills and some has both basic as well as advanced skills. The metrics used for assessing can also vary significantly between the simulators. Not all simulators have force feedback. Moreover, fidelity of these simulators is also very important (ie., whether they have undergone construct validity). These issues greatly limit the usefulness of metal analysis since the number of articles were going to be always very few.

The authors have summarized the results from the 7 papers but my concern is that the conclusion is severely limited by the factors mentioned above. Unlike the authors mentioned in the paper, there has been a review published on the virtual reality training for laparoscopic surgery (Alaker M, Wynn GR, Arulampalam T. Virtual reality training in laparoscopic surgery: A systematic review & meta-analysis. Int J Surg. 2016 May;29:85-94.). Also, the authors have cited the paper by Nagendran et al., which does a meta analysis of virtual reality training in laparoscopic surgery with focus on randomized clinical trials with primary and secondary outcomes but failed to elaborate on it. In view of the two review papers that has been published and the current review paper which by exclusion criteria doesn’t include papers beyond 2015, the real impact of the metal analysis is minimal. At the least, the authors should compare the conclusions from these two meta analysis to their conclusions and explain clearly any discrepancies.

Thank you for pointing out this review from Alaker M et al. (2016). Indeed, we agree that there was a need to go more in depth in the existing literature. In order to address this issue, we have expanded the section of our introduction in which we discuss the standing of our review in relation to the ones done prior. Furthermore, as you suggested, other changes have been made in our discussion, considering the results of our review and how they compare to the previous ones. We believe that our paper has significantly improved with these adjustments, your feedback was much appreciated.

The authors mention that in the article by Diesen et al., they didn’t report the VRS used in the study, but they did mention it in the discussion section (3rd paragraph). They have used both the Medical Education Technologies Inc. laparoscopic simulator as well as the Immersion virtual reality laparoscopic simulator. One would expect a thorough review of the paper especially for meta analysis. This needs to be fixed if the authors decided to revise the paper.

You are right, we have updated this in our review and made a thorough revision of our paper for any other misplaced information.

Reference 13 is incomplete and is also listed as part of reference 12.

This has been fixed and now both references 12 and 13 are complete.

Reviewer #2: I believe that this is an important piece. It is absolutely shocking that after 15 years of marketing and pushing for laparoscopic simulation the most we can find in the literature at this point to support its use is 7 papers. The number of poorly constructed C and D level publications in this space is simply incredible. The sections of the paper breaking down these shortcomings is a call-to-arms for those in this space to structure their research correctly and to power the studies appropriately. Thank you. There are a few grammatical issues in the paper that should be reviewed; a number of past/present tense inconsistencies as well as some simple capitalizations in the middle of sentences, etc.

We agree with you and hope that our research can play its role in helping the scientific community. Additionally, our paper was sent for grammatical correction to a british translator.

---

## [Decision Letter · Decision Letter 1]

26 Apr 2021

PONE-D-20-37286R1

The development of laparoscopic skills using virtual reality simulations: a systematic review

PLOS ONE

Dear Dr. Meyer,

Thank you for submitting your manuscript to PLOS ONE. After careful consideration, we feel that it has merit but does not fully meet PLOS ONE’s publication criteria as it currently stands. Therefore, we invite you to submit a revised version of the manuscript that addresses the points raised during the review process. As you can see, the article needs improvement, especially, in the light of discussion of how the findings compare to earlier research,

We look forward to receiving your revised manuscript.

Kind regards,

Mohammed Saqr, Ph.D

Academic Editor

PLOS ONE

Journal Requirements:

Reviewers' comments:

Reviewer's Responses to Questions

**Comments to the Author**

1. If the authors have adequately addressed your comments raised in a previous round of review and you feel that this manuscript is now acceptable for publication, you may indicate that here to bypass the “Comments to the Author” section, enter your conflict of interest statement in the “Confidential to Editor” section, and submit your "Accept" recommendation.

Reviewer #1: (No Response)

Reviewer #3: All comments have been addressed

2. Is the manuscript technically sound, and do the data support the conclusions?

Reviewer #1: Partly

Reviewer #3: Yes

3. Has the statistical analysis been performed appropriately and rigorously? 

Reviewer #1: N/A

Reviewer #3: I Don't Know

4. Have the authors made all data underlying the findings in their manuscript fully available?

Reviewer #1: No

Reviewer #3: Yes

5. Is the manuscript presented in an intelligible fashion and written in standard English?

Reviewer #1: Yes

Reviewer #3: Yes

6. Review Comments to the Author

Reviewer #1: The authors have addressed some of the issues that were noted in their initial submission. Specifically, added reference to a relevant meta-analysis paper by Alaket et al. With respect to new information that can be obtained from this meta analysis, compared to the other paper even though there is a clear four years difference, is very minimal. Even more, in this paper there is no meta-analysis in the end which is a big let down and results were more a qualitative summary. Discussion of cost of simulators is perfectly fine but over time, VR simulator can save cost for the hospital by reducing or eliminating the need for proctors, no need to replenish materials such has synthetic, cadaveric or animal tissues or organs etc. (advantages of VRS). I think comparing to the existing meta-analysis, this review paper doesn't add more information and may be premature at this point for publication unless criteria of selection is changed to be able to perform an actual meta-analysis.

Reviewer #3: This systematic review article summarizes the results of using virtual reality simulators for developing laparoscopic skills from1529, of which, only 7 were selected. The authors concluded that virtual reality simulator has no significant edge over traditional training. I believe that this topic is important, and such review is needed.

Despite not including studies beyond 2015, the authors stated that no eligible studies were found. I belive this point should be highlighted in the abstract.

I also advise the authors to update their manuscript with the following references :

1- Gurusamy K, Aggarwal R, Palanivelu L, Davidson BR. Systematic review of randomized controlled trials on the effectiveness of virtual reality training for laparoscopic surgery. British Journal of Surgery. 2008 Sep;95(9):1088-97.

2- Jin C, Dai L, Wang T. The application of virtual reality in the training of laparoscopic surgery: a systematic review and meta-analysis. International Journal of Surgery. 2020 Dec 9.

Finally, I would also suggest that the paper be reviewed by a native English speaker to adjust for some grammatical errors.

7. PLOS authors have the option to publish the peer review history of their article (what does this mean?). If published, this will include your full peer review and any attached files.

Reviewer #1: No

Reviewer #3: No

---

## [Author Response · Author response to Decision Letter 1]

7 May 2021

Dear reviewers,

Thank you for inviting us to submit a revised draft of our manuscript.

We also appreciate the time and effort you have dedicated to providing insightful feedback on ways to strengthen our paper. Thus, it is with great pleasure that we resubmit our article for further consideration. We have incorporated changes that reflect the detailed suggestions you have provided. We also hope that our edits and the responses we provide below satisfactorily address all of the issues and concerns you have previously noted.

Reviewer #3: This systematic review article summarizes the results of using virtual reality simulators for developing laparoscopic skills from1529, of which, only 7 were selected. The authors concluded that virtual reality simulator has no significant edge over traditional training. I believe that this topic is important, and such review is needed.

Despite not including studies beyond 2015, the authors stated that no eligible studies were found. I belive this point should be highlighted in the abstract.

I also advise the authors to update their manuscript with the following references:

1- Gurusamy K, Aggarwal R, Palanivelu L, Davidson BR. Systematic review of randomized controlled trials on the effectiveness of virtual reality training for laparoscopic surgery. British Journal of Surgery. 2008 Sep;95(9):1088-97.

2- Jin C, Dai L, Wang T. The application of virtual reality in the training of laparoscopic surgery: a systematic review and meta-analysis. International Journal of Surgery. 2020 Dec 9.

Finally, I would also suggest that the paper be reviewed by a native English speaker to adjust for some grammatical errors.

Indeed, we do believe the abstract should inform our readers about the lack of eligible studies beyond 2015, as this data is relevant and they could benefit from it beforehand. Hence we have incorporated your suggestion within the abstract limitations section.

Also, thank you for pointing out these two references, as they significantly improved our research and have been included within our text. Finally, our paper was once again revised by another native english-speaking translator and we hope all grammatical issues have now been corrected.

Reviewer #1: The authors have addressed some of the issues that were noted in their initial submission. Specifically, added reference to a relevant meta-analysis paper by Alaket et al. With respect to new information that can be obtained from this meta analysis, compared to the other paper even though there is a clear four years difference, is very minimal. Even more, in this paper there is no meta-analysis in the end which is a big let down and results were more a qualitative summary.

We have considered all your suggestions and really appreciate the consideration for our research, thank you for providing us with another significant point of view, which inspired us to seek improvements for elevating the quality of our research. We hope now that our paper meets your expectations with the newest changes and improvements, just as we expect that some of our decisions regarding other changes you proposed have been properly explained

Discussion of cost of simulators is perfectly fine but over time, VR simulator can save cost for the hospital by reducing or eliminating the need for proctors, no need to replenish materials such has synthetic, cadaveric or animal tissues or organs etc. (advantages of VRS).

Although the main goal of our review was not to analyse the cost-effectiveness of VRS, we believe that such considerations are of great importance and, as such, agree on the necessity to explore this subject. As you suggested, there was a need to expand our discussion on the matter of advantages of VRS and we, accordingly, addressed it. We now hope this segment of our review pleases and offers a more appropriate understanding of the question at hand.

I think comparing to the existing meta-analysis, this review paper doesn't add more information and may be premature at this point for publication unless criteria of selection is changed to be able to perform an actual meta-analysis.

Regarding the changes in eligibility criteria you suggested, we believe that, given the current stage of review, such changes would be unviable, as they would either demand a complete do over of the entire project or be included in such a late stage, as to compromise the impartiality of the paper. It should also be noted that unlike Alaker et al., the other review you presented, we opted for a more restrictive eligibility criteria that explicitly required both pre and post-assessment for every study we included, in order to favor quality over quantity.

In the near future, we hope to once more submit other projects and studies to this Journal, making sure to take into account all suggestions you made early on and, certainly, your remarks will inspire us to continue seeking the betterment of any paper we may produce.

---

## [Decision Letter · Decision Letter 2]

19 May 2021

The development of laparoscopic skills using virtual reality simulations: a systematic review

PONE-D-20-37286R2

Dear Dr. Meyer,

We’re pleased to inform you that your manuscript has been judged scientifically suitable for publication and will be formally accepted for publication once it meets all outstanding technical requirements.

Kind regards,

Mohammed Saqr, Ph.D

Academic Editor

PLOS ONE

Additional Editor Comments (optional):

Reviewers' comments:

Reviewer's Responses to Questions

**Comments to the Author**

1. If the authors have adequately addressed your comments raised in a previous round of review and you feel that this manuscript is now acceptable for publication, you may indicate that here to bypass the “Comments to the Author” section, enter your conflict of interest statement in the “Confidential to Editor” section, and submit your "Accept" recommendation.

Reviewer #3: All comments have been addressed

2. Is the manuscript technically sound, and do the data support the conclusions?

Reviewer #3: Yes

3. Has the statistical analysis been performed appropriately and rigorously? 

Reviewer #3: I Don't Know

4. Have the authors made all data underlying the findings in their manuscript fully available?

Reviewer #3: Yes

5. Is the manuscript presented in an intelligible fashion and written in standard English?

Reviewer #3: Yes

6. Review Comments to the Author

Reviewer #3: All of my comments have been addressed by the authors. No further revisions are needed from my side.

7. PLOS authors have the option to publish the peer review history of their article (what does this mean?). If published, this will include your full peer review and any attached files.

Reviewer #3: **Yes: **Ismail Ibrahim Ismail

---

## [Editor Report · Acceptance letter]

24 May 2021

PONE-D-20-37286R2 

The development of laparoscopic skills using virtual reality simulations: a systematic review 

Dear Dr. Meyer:

I'm pleased to inform you that your manuscript has been deemed suitable for publication in PLOS ONE. Congratulations! Your manuscript is now with our production department. 

Kind regards, 

on behalf of

Dr. Mohammed Saqr 

Academic Editor

PLOS ONE